# Modeling and Molecular Dynamics of the 3D Structure of the HPV16 E7 Protein and Its Variants

**DOI:** 10.3390/ijms22031400

**Published:** 2021-01-30

**Authors:** Ciresthel Bello-Rios, Sarita Montaño, Olga Lilia Garibay-Cerdenares, Lilian Esmeralda Araujo-Arcos, Marco Antonio Leyva-Vázquez, Berenice Illades-Aguiar

**Affiliations:** 1Laboratorio de Biomedicina Molecular, Facultad de Ciencias Químico-Biológicas, Universidad Autonóma de Guerrero, Chilpancingo, CP 39087, Mexico; ciresthelbello@uagro.mx (C.B.-R.); olgaribayce@conacyt.mx (O.L.G.-C.); esme_1018@hotmail.com (L.E.A.-A.); leyvamarco13@gmail.com (M.A.L.-V.); 2Laboratorio de Bioinformática y Simulación Molecular, Facultad de Ciencias Químico Biológicas, Universidad Autónoma de Sinaloa, Culiacán Sinaloa, CP 80030, Mexico; 3CONACyT-Universdad Autónoma de Guerrero, Chilpancingo, CP 39087, Mexico

**Keywords:** HPV, E7, variants, molecular dynamics simulation

## Abstract

The oncogenic potential of high-risk human papillomavirus (HPV) is predicated on the production of the E6 and E7 oncoproteins, which are responsible for disrupting the control of the cell cycle. Epidemiological studies have proposed that the presence of the N29S and H51N variants of the HPV16 E7 protein is significantly associated with cervical cancer. It has been suggested that changes in the amino acid sequence of E7 variants may affect the oncoprotein 3D structure; however, this remains uncertain. An analysis of the structural differences of the HPV16 E7 protein and its variants (N29S and H51N) was performed through homology modeling and structural refinement by molecular dynamics simulation. We propose, for the first time, a 3D structure of the E7 reference protein and two of Its variants (N29S and H51N), and conclude that the mutations induced by the variants in N29S and H51N have a significant influence on the 3D structure of the E7 protein of HPV16, which could be related to the oncogenic capacity of this protein.

## 1. Introduction

Cervical cancer (CC) is a serious public health problem worldwide [1]. Human papillomavirus (HPV) infection is the main cause of CC [2]. However, not all infections cause tumor development due to the oncogenic potential of different types of HPV [2,3,4,5]. For instance, high-risk HPV types 16 and 18 are reported with the greatest frequency in cervical cancer [4,6,7]. The oncogenic potential of high-risk HPV is due to E6 and E7 oncoprotein production, which initiates a series of alterations associated with cell transformation through the inactivation of p53 and pRB, respectively [4,8,9]. The E7 protein plays a central role in the life cycle of the virus and tumor transformation [10,11,12,13]. The E7 protein has the ability to transform due to its interaction with multiple targets involved in different cellular processes, such as apoptosis, angiogenesis, cell immortality, resistance to cell death, and Mesenchymal Epithelial Transition (EMT) [14,15,16,17,18].

The E7 protein of HPV16 is composed of 98 amino acids and is divided into three conserved regions, CR1, CR2, and CR3 [19,20,21]. CR1 of E7 consists of residues M1–L15 in the N-terminal region. CR2 (Q16–I38) contains the LXCXE motif, which establishes high-affinity interactions with the retinoblastoma protein (pRB) [21,22,23]. In addition, E7 contains a recognition motif for casein kinase II (CKII), which phosphorylates serine residues 31 and 32. These phosphorylation events confer structural stability upon the protein [23,24,25]. Finally, CR3 (D39-P98), located in the C-terminus, contains two CXXC motifs that form a zinc finger structure involved not only in protein stability, but also in its dimerization [19,20,26,27,28,29].

The three-dimensional structure of the E7 protein allows it to associate with cell cycle regulatory proteins [23,24,25,30]. The N-terminus confers conformational plasticity to promote interactions with other proteins. These transition properties and the plasticity of E7 play key roles in its ability to interact with different components of cellular processes and thus initiate malignant transformation [2,19,28,31,32].

It has been described that mutations in the E7 gene of HPV16 are associated with viral pathogenicity; this is attributed to intratypical variations of these mutations that may induce a residue change in the sequence. At least 23 variants have been reported to induce amino acid changes in the protein. These changes are epidemiologically related to the oncogenicity of the virus in different populations [33,34,35,36], where variants G647 (N29S) and A712 (H51N) have been found to be most frequently associated with CC [33,35,37,38]. However, only the N29S variant has been related to a significant oncogenic potential due to an additional phosphorylation site compared to the E7 reference [39]. Experimental and epidemiological evidence suggests that the effects caused by the particular mutation of each variant could generate a change in the 3D structure, modifying the interaction with target proteins to induce their oncogenic potential [33,34,35,37,38,39,40]. Therefore, the objective of this study was to elucidate the structural changes in the E7 reference protein and its variants by using structural analysis *in silico* and molecular dynamics (MD) simulations.

## 2. Results and Discussion

### 2.1. Modeling and Structural Alignment of the E7 Reference and Its Variants

Multiple alignments of the sequences were performed to observe the residue changes of the variants with respect to the E7 reference. Figure 1A shows that the changes in the N29S variant (blue) were located inside the CR2 region, and those of the H51N variant (orange) were in the CR3 region. The 2D structures of the E7 reference and variants were predicted and were identified via alignment (green line); the results show that the secondary structure was not changed between the reference and the variants.

To predict the 3D model of the E7 reference, we used the crystal structure of the C-terminal domain of the E7 HPV45 protein (PDB ID: 2EWL) as a template in the I-TASSER server. Figure 1B shows the structural alignment between 2EWL (pink) and the 3D model of the E7 reference (green). After obtaining the E7 model, we generated the variants with their respective mutations N29S (orange) and H51N (blue) in PyMOL, as seen in Figure 1C. The E7 reference contains a Zn finger. To place the Zn finger of the CR3 region in E7, we first deprotonated the cysteine residue with the CYS_D patch, and, subsequently, the ZN_C structural patch was added to the Zn finger at C 58, 61, 91, and 94. These proteins have phosphorylation sites, and we used the SP2 structure patch to place the phosphorylation sites of S31 and 32 (in the reference and H51N) and S29, 31, and 32 (in N29S). Figure 1D shows the phosphorylation and Zn fingers.

### 2.2. N29S Presents the Most Stable Trajectory Compared to the E7 Reference and H51N

We performed an MD simulation of 200 ns to understand the effect of the mutations on the structural motion and stability of the E7 reference and the N29S and H51N variants. Figure 2A shows changes in protein stability, which were evaluated using the root mean square deviation (RMSD). The E7 reference (green) reached equilibrium after 80 ns. N29S (orange) reached equilibrium after 20 ns and remained stable throughout the trajectory simulation. In contrast, H51N (blue) reached equilibrium after 20 ns and remained stable until 100 ns, but in the last 100 ns of the trajectory simulation, we observed an equilibrium period from 100 to 150 ns and then an oscillating pattern of approximately 2 Å. This behavior of the variant may be due to the nature of the residue change. To understand how the mutations affected protein backbone motion, the root mean square fluctuation (RMSF) was calculated. RMSF was used to explore the flexibility of the Cα of a protein throughout the trajectory of the MD simulation. Higher RMSF values indicated greater flexibility during the MD simulation. As Figure 2B indicates, the principal peak of fluctuation was located between residues N29 and H51 (shaded in pink and violet) in all the structures. However, this fluctuation was highest for the H51N variant, and the fluctuation of N29S was minor with respect to the E7 reference. The principal peak of fluctuation into the CR2 and CR3 regions of these proteins, where structural differences were observed. The radius of gyration (Rg) is the parameter that defines the balance of the conformations of the total system in terms of compaction with respect to the folding and unfolding of the proteins throughout the simulation. The Rg between the E7 reference (green) and N29S (blue) showed similar compaction during dynamics, while the H51N (blue) variant exhibited an expansion in the final 50 ns of the trajectory simulation. This behavior agreed with the RMSD and RMSF values, showing that H51N had the highest peak of fluctuation among the proteins. The N29S variant showed a more compact structure throughout the MD simulation than the E7 reference and the H51N variant, as seen in Figure 3C. This behavior on the trajectory of N29S may suggest the importance of the extra phosphorylation site on S29.

To validate the quality of the 3D structure of the E7 reference and variants, we generated a Ramachandran plot. Figure 3A depicts the results of the E7 reference: 85.3% of the amino acids were in the favored region, 7.1% were in the allowed region, and 3.6% were in the not-allowed region. The N29S variant, Figure 3B, showed that 88.1% of the residues were in the favored region, while 10.7% were in the allowed region, and 1.2% were in the not-allowed region. With respect to the H51N variant in Figure 3C, 90.8% of its amino acids were found in the favored region, 8.0% in the allowed region, and 1.1% in the not-allowed region. These data show that the refined models are reliable. To observe the structural changes between reference E7 and the variants, an alignment of the average structure of the reference E7 and the N29S and H51N variants was performed. The alignment in Figure 3D shows that there were differences in the 3D structure between the proteins. While changes were observed in the three regions of the proteins, the main changes were found in the CR1 and CR2 regions, as demonstrated in Appendix A, which contained the main motifs that enabled E7 to interact with its molecular targets. These changes may explain the different oncogenic potentials of the E7 variants.

### 2.3. H51N Loses β-Sheets throughout the Trajectory Simulation

The E7 protein has a disordered N-terminus that includes the CR1 and CR2 domains, which are involved in most of the protein interactions with its molecular targets [40,41]. To understand the differences between the E7 reference and its variants, we explored the changes in the secondary structure of the protein throughout the MD simulation. The most significant changes in the 2D structures were in the E7 reference, because it lost the α-helix and the β-sheets, as can be seen in Figure 4A. However, Figure 4B indicates that most of the secondary structures of N29S were preserved. The H51N variant conserved the α-helix but lost the β-sheets after 140 ns, shown in Figure 4C. The loss of the β-sheet structures may explain the oscillating behavior observed in the final 50 ns, as indicated by the RMSD values and the compaction of the Rg. This outcome revealed that the mutation made the N29S variant more structurally stable than the E7 reference and H51N. These findings agree with the trajectory analysis, where N29S had the most stable RMSD, the lowest RMSF, and major compaction compared to the others. 

### 2.4. Surface Electrostatic Potential of E7 and Its Variants

The LxCxE motif of the CR2 region of E7 is crucial for the interaction with pRB, which is its main molecular target. Studies have shown that mutations in the E7 LxCxE motif drastically influence pRB degradation [19,42]. Our results demonstrated an essential structural change induced by mutations. To determine whether the LxCxE motif of E7 was different between the systems, an analysis of the electrostatic potential was performed using the adaptive Poisson-Boltzmann solver (APBS) plug-in in PyMOL to map the electrostatic potential of the surface of the E7 reference and its variants, shown in Figure 5. The charge differences of the variants with respect to the E7 reference were noticeable. The E7 reference had a negative charge on the LxCxE motif (inside the green circle), while the N29S variant had not only a prominent negative charge, but also an electrostatically positive region and a neutral region. In the H51N variant, the LxCxE motif appeared to be maintained by a combination of hydrophobic interactions and electrostatic complementation, as it contained negative, neutral, and positive regions. These differences may be related to the affinity of their interactions with target proteins and explain the differences in the oncogenic potential between them.

### 2.5. The CR3 Region Allows Homodimer Formation of the E7 Protein

It has been reported that the E7 protein can form homodimers, allowing interactions with more than one protein at the same time and thus enabling the formation of protein complexes [23,27,29], and that the amino acids in CR3 of E7 are involved in homodimer formation [13,22,23,29]. Our results showed that E7-E7 homodimer formation involved binding in the CR3 regions, leaving the N-terminal regions free for interactions with other proteins, indicated in Figure 6. This finding agrees with those experimentally reported.

### 2.6. Principal Component Analysis (PCA)

PCA was performed to support the results of the MD simulation and to understand the structure and conformational changes of the E7 reference, N29S, and H51N by calculating the atomic fluctuation covariance matrix, shown in Figure 7A. The first 20 main components represented 72–75% of the total movement (72.5%, 73.9%, and 74.3% for the E7 reference, N29S, and H51N, respectively). This analysis suggests that the properties of the movements described by the first PCs were different in the three protein systems, as seen in Figure 7B. The projection of PC1 and PC2 for each of the proteins (E7 reference, N29S, and H51N), as seen in Figure 7C–E, showed that the H51N variant presented more significant displacement, covering a wider range of space than the E7 reference and the N29S variant. The N29S variant had the most restricted movement, making it the most stable of the three protein systems. This result showed differential distribution between the protein systems, suggesting that the conformational mobility changed because of the mutation of a single residue; see Figure 7D,E.

## 3. Materials and Methods

### 3.1. The 3D Structure of E7 and Its Variants

Multiple alignments of the sequences and the calculations of the percent identity of the amino acid sequences were performed using CLUSTAL X 1.81 [43]. The secondary structure of the E7 protein models and its variants were predicted using PSIPRED server [44], and templates were located in the Protein Data Bank (PDB) (https://www.rcsb.org/). The HPV16 E7 structure was built by the I-TASSER server [42] using the P03129 sequence obtained from the UniProtKB database (http://www.uniprot.org). The homology model was generated using the structure of the C-terminal portion of the E7 oncoprotein of HPV45 as the template (PDB ID: 2EWL) [20] in the I-TASSER server. After selecting the most energetically stable models, we evaluated their quality by generating Ramachandran’s plots with the RAMPAGE server (http://mordred.bioc.cam.ac.uk/~rapper/rampage.php). To build the N29S and H51N variants, the 3D structure obtained from the PDB was edited with the PyMOL program (http://www.pymol.org). The alignment of the 3D structures of the models by homology and the refined models were performed with the alignment plug-in of the PyMOL program and visualized with VMD software [45].

### 3.2. Molecular Dynamics Simulation

For the preparation of the molecular dynamics simulation, a collaboration was carried out with the Laboratory of Molecular Modeling and Bioinformatics of the Faculty of Biological Chemical Sciences at the Autonomous University of Sinaloa using the Hybrid Cluster “Xiuhcoatl” in CGSTIC-CINVESTAV (http://clusterhibrido.cinvestav.mx). The proteins were prepared with CHARMM/VMD NAMD software [45,46,47]. The E7 protein contains a Zn finger motif and is phosphorylated; specific structural patches were used in each case using CHARMM. The CHARMM 36 force field was used to create the protein topology file [48]. To neutralize the system, we added 4507 water molecules and 35 Na+ and 12 Cl^−^ to the E7 reference. To the N29S variant, we added 4578 water molecules, 35 Na^+^, and 13 Cl^−^, and to the H51N variant, 4731 water molecules, 33 Na^+^, and 13 Cl^−^ were added. Each system was solvated in a cubic box of TIP3P water with a minimum distance of 10 Å imposed between the solute atoms and the edge of the box. The system charges were balanced by adding the chloride and sodium ions required for each protein. The system was minimized to 10,000 steps followed by an equilibrium for 50,000 steps at a constant temperature and pressure using NPT ensemble at 300 K for one ns with protein atoms restrained. Then, 200-ns-long MD simulations were run for each protein (E7 and its variants), considering all proteins as solubles, without position restraints under periodic boundary conditions. The temperature was maintained employing Langevin dynamics, while the pressure was kept constant at one bar using Nosé-Hoover-Langevin piston [49,50]. The time step was set to 2.0 fs, and the coordinates were saved for analyses every one ps.

To visualize and analyze the results of the simulation, we used the VMD graphics program [45].

### 3.3. Trajectory and Secondary Structure Analysis

The stability and conformational changes of the trajectory for E7 and its variants were evaluated by analyzing the RMSD, the RMSF, and the Rg. The dynamics data were calculated by the Carma program [51] and plotted in the Excel program. The secondary structure calculation was carried out in the VMD program using the timeline plug-in through the entire trajectory simulation of 200 ns.

### 3.4. Calculation of Surface Electrostatic Potential

The electrostatic potential energy was calculated using the APBS package in PyMOL. This plug-in solves the Poisson-Boltzmann equation for biomolecules. The potential areas were colored using −5 and +5 kT/e, where k is the Boltzmann constant, T is the temperature, and e is the electron charge. The figures were visualized and generated with the PyMOL program.

### 3.5. Homodimer Prediction

To predict the homodimer formation of the E7 reference, the COTH server was employed (https://zhanglab.ccmb.med.umich.edu/COTH/) [52], and the complex with the highest score was selected.

### 3.6. Principal Component Analysis

The goal of protein simulation is to generate enough configurations of the system of interest to identify functionally relevant movements. Principal component analysis (PCA) has been widely used to reduce the complexity of the data obtained from MD simulation trajectories by recovering the collective movement of atoms in the simulated trajectories that are essential for biological processes. PCA was performed with the data extracted from a 200-ns MD simulation. The projected eigenvectors and eigenvalues obtained during the first two principal components (PC1 and PC2) runs were calculated with the Carma program. The dPCA was calculated considering only the Cα of the protein.

## 4. Conclusions

HPV has been described as the principal cause for the development of cervical cancer, and this potential is due to the production of its oncoproteins. One of the most important is the E7 oncoprotein, which disrupts the cell cycle through direct interactions with regulatory proteins during this cellular process. This protein presents intratypical variants that have been epidemiologically related to the oncogenic potential of this virus. While there have been no reports about the effect of these variations in the structure of the protein, these effects may be closely related to the oncogenic differences present in the variants.

Our findings show, for the first time, that an amino acid change in a variant (N29S and H51N) affected the 3D structure of the E7 protein. These changes translated into increased or reduced stability of the 3D structure of the protein. The H51N variant showed a lower 3D structure stability than the other systems. This could be explained by the nature of the amino acid charge: since H has a positive charge, while N is an uncharged amino acid, the heterogenicity found in the charges on the surface of this variant could, as a consequence, affect the binding with the pRB protein; however, this must be supported with more studies. Interestingly, the N29S mutation confered greater 3D structure stability compared to those of the H51N variant and the E7 reference; this outcome may be related to the fact that N29S is an additional phosphorylation site. The stability we observed in the MD simulation trajectories of N29S could be induced by this additional phosphorylation, which affected the global structure of the protein. Experimental studies have shown that the phosphorylation of the E7 protein is related to an increased half-life and a greater capacity for protein transformation [53,54]. Thus, phosphorylation may explain why this protein showed greater stability in the MD simulation trajectory. The mutation of asparagine 29 to serine in the N29S variant was reported to confer an additional phosphorylation site at S29 due to its proximity to the CKII recognition motif, which phosphorylates S31 and S32. It has been reported that S29 phosphorylation is involved in the increase in the number and size of colonies with respect to the E7 reference, which indicates an improvement in the oncogenic potential of this variant [39]. These conformational changes induced by mutations may explain the difference between their oncogenic potential. However, the effect of these structural changes on the interaction with the molecular targets of the E7 protein remains to be elucidated.

## Figures and Tables

**Figure 1 ijms-22-01400-f001:**
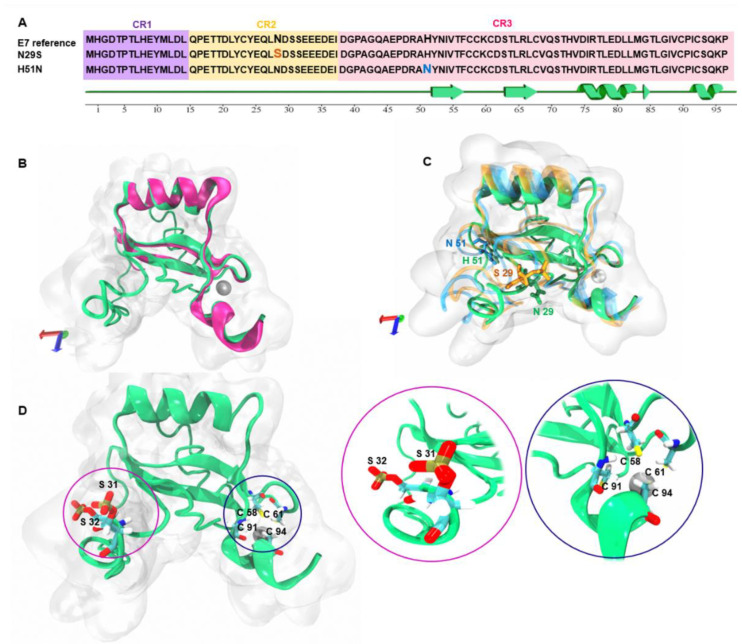
Amino acid sequence alignment and 3D model of E7 and variants. (**A**) The amino acid sequence of HPV16 E7 (P03129) is aligned with the sequences of the N29S and H51N variants. Regions highlighted in orange and blue indicate the amino acid change in each variant. (**B**) 3D structure of the E7 reference (green) and HPV45 crystal template (PDB ID: 2EWL) (pink). (**C**) 3D structural alignment of the E7 reference (green), N29S (orange), and H51N (blue) variants. (**D**) Phosphorylation sites at serines 31 and 32 in the N-terminal region (pink circle) and the presence of Zn finger in cysteines 58, 61, 91, and 94 at the C-terminal (blue circle); the zoom of these regions are next to Figure D.

**Figure 2 ijms-22-01400-f002:**
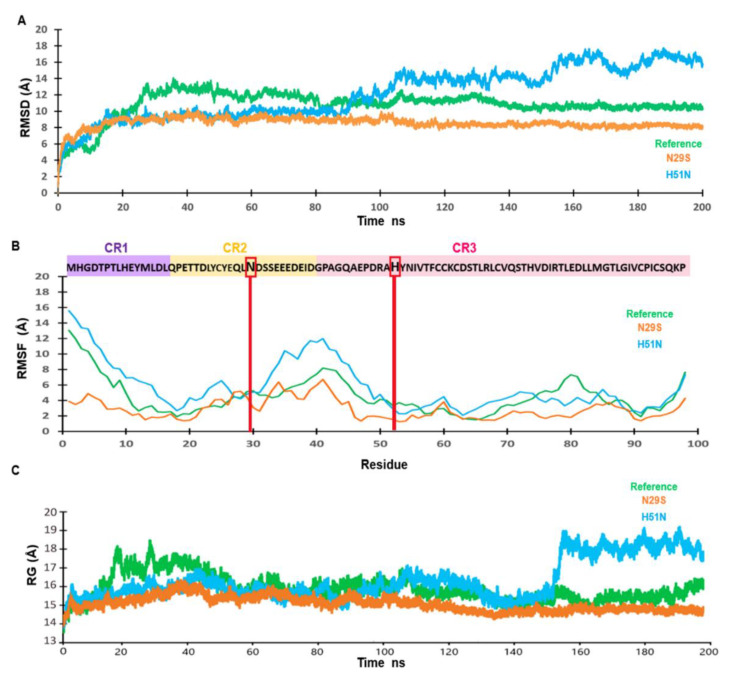
Trajectory analysis of molecular dynamics (MD) simulation of E7 and variants. (**A**) The root mean square deviation (RMSD). The trajectory of the systems is shown after 200 ns of MD simulation. (**B**) The root mean square fluctuation (RMSF) after 200 ns of MD simulation. In the upper panel, the E7 reference amino acid sequence (purple shading) are the amino acids from the CR1 region. In CR2 (yellow) and CR3 (pink), the red lines and boxes indicate the mutations of each variant. (**C**) Radius of gyration (Rg) of the E7 reference (green), N29S (orange), and H51N (blue).

**Figure 3 ijms-22-01400-f003:**
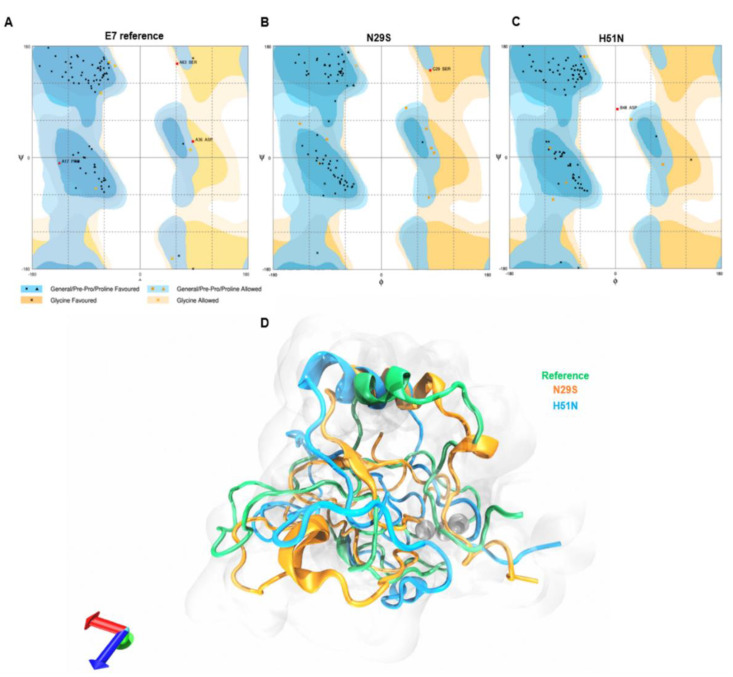
Evaluation and structural alignment of the average structure of E7 reference and variants. (**A**) Ramachandran plot analysis of E7 reference. (**B**) Ramachandran plot analysis N29S. (**C**) Ramachandran plot analysis H51N. (**D**) Structural alignment of the average structures of the E7 reference (green), N29S (orange), and H51N (blue).

**Figure 4 ijms-22-01400-f004:**
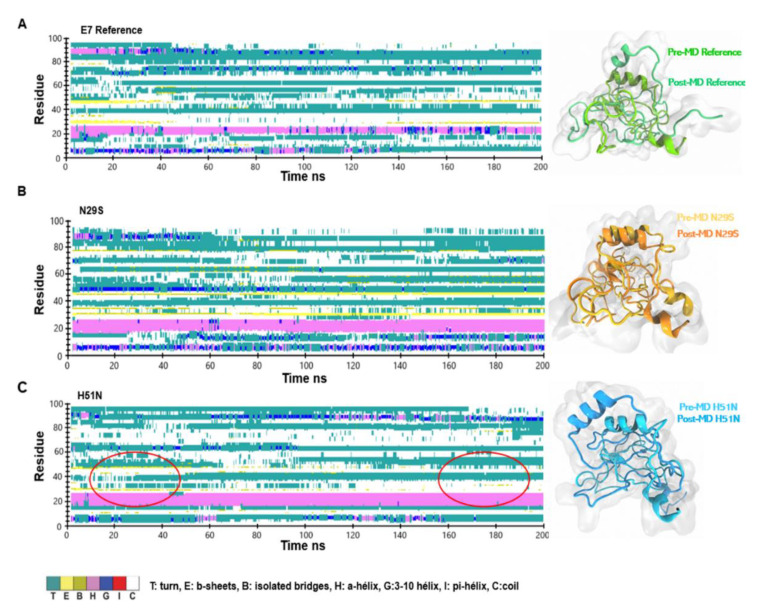
Analysis of the secondary structure through 200 ns of molecular dynamics (MD) simulation of the proteins. (**A**) Secondary structures analysis of the E7 reference. Right: Structural alignment of the initial structure and snapshot at 200 ns for E7 reference. (**B**) Secondary structure analysis of the N29S variant. Right: Structural alignment of the initial structure and snapshot corresponding to 200 ns for N29S. **C**) Secondary structure analysis of the H51N variant. The principal zones of β-sheets are indicated with circles in red. Right: Structural alignment of the initial structure and snapshot belonging to 200 ns for H51N.

**Figure 5 ijms-22-01400-f005:**
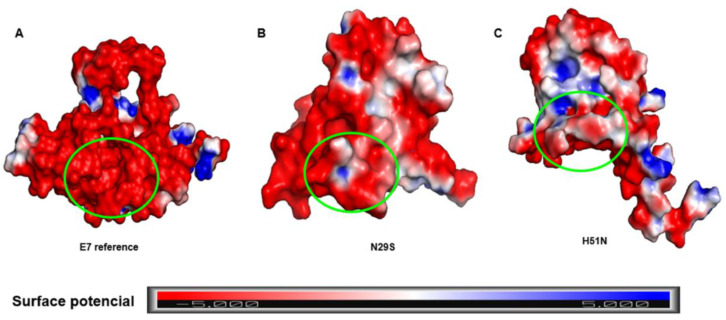
Surface representation of the electrostatic potential of the E7 protein and variants. (**A**) E7 reference. (**B**) N29S variant. (**C**) H51N variant. The surface colors are represented in red (−5) or blue (+5). The marked regions inside the green circle show the different charge distributions in the LxCxE motif.

**Figure 6 ijms-22-01400-f006:**
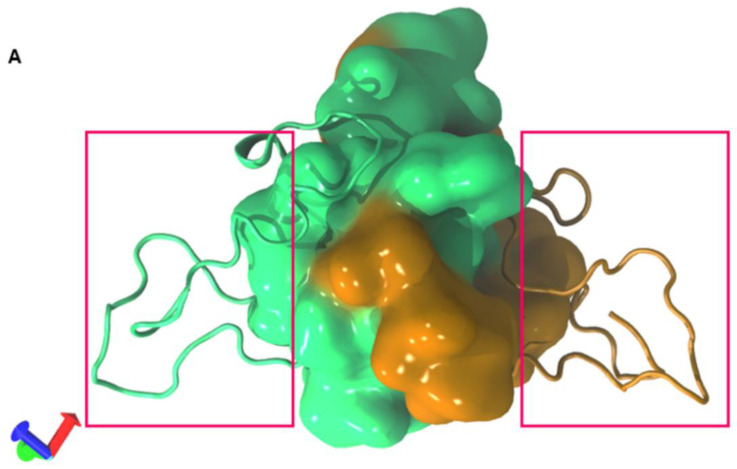
Homodimer prediction of E7. (**A**) Homodimer of the reference E7 average structure model. The CR3 region of each monomer is represented in quicksurf (green and orange). In the boxes, the N-terminal region of each monomer is highlighted.

**Figure 7 ijms-22-01400-f007:**
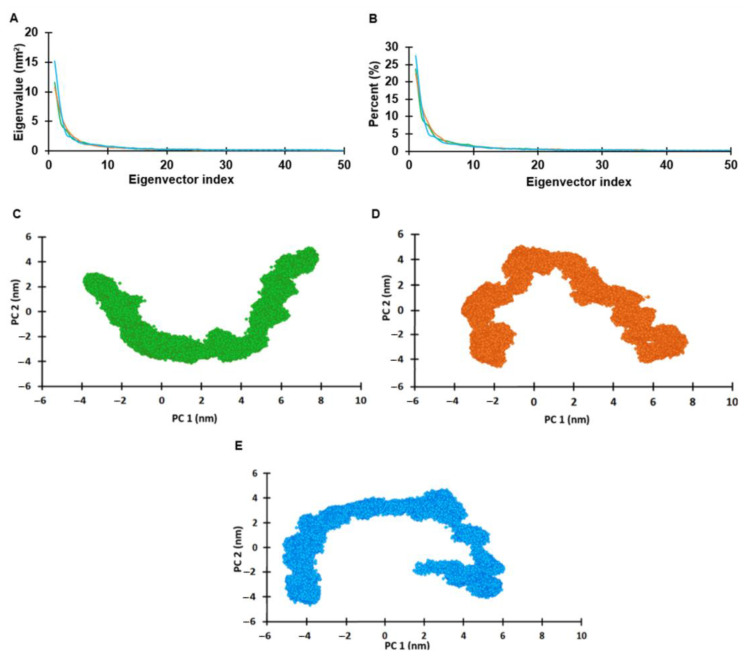
Principal components analysis of the E7 protein and its variants. (**A**) Eigenvectors of the covariance matrix. (**B**) Percentage of each eigenvector vs. eigenvalues. (**C**) Projection of the movement in the phase space between the first and second eigenvectors (PC1 vs. PC2) of the reference E7. (**D**) Projection of the movement in the phase space along the first and second eigenvectors (PC1 vs. PC2) of N29S. (**E**) Projection of the movement in the phase space along the first and second eigenvectors (PC1 vs PC2) of H51N. Reference E7 protein is represented in green, N29S orange, and H51N in blue.

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
