# Peer review of "Modeling and Molecular Dynamics of the 3D Structure of the HPV16 E7 Protein and Its Variants"

_ijms, 2021, doi:10.3390/ijms22031400_

Round 1
Reviewer 1 Report
Authors report an attempt to investigate the structure of E7 protein and its variants by means of computational biochemistry. The results, in general, seem to be backed by the modeling performed. Nevertheless, they say little about the mechanism of E7 protein variants oncogenicity. It would be better to explore more and add these data to paper.
Reviewer 2 Report
Paper by Ciresthel Bello-Rios et all presents the results of molecular dynamics simulation of HPV16 E7 protein and its two variants. Authors show how modifications of E7 reference protein change its 3D structure. The presented work is well written, however, there is a few methodological information missing.
I recommend publication after minor revision regarding details of molecular dynamics simulation set up.
Line 135 - What is RG - is its radius of gyration? If so unify the notation
What thermostat and barostat did you use for MD simulations?
What boundary conditions were applied?
What was the time step?
Was the system with water equilibrated before the simulation?
